# Microbial Crosstalk with Therapy: Pharmacomicrobiomics in AML—One Step Closer to Personalized Medicine

**DOI:** 10.3390/biomedicines13071761

**Published:** 2025-07-18

**Authors:** Aneta Nowicka, Hanna Tomczak, Edyta Szałek, Agnieszka Karbownik, Lidia Gil

**Affiliations:** 1Department of Hematology and Bone Marrow Transplantation, Poznan University of Medical Sciences, 60-569 Poznan, Poland; lidia.gil@skpp.edu.pl; 2Faculty of Medicine and Health Sciences, Calisia University, 62-800 Kalisz, Poland; hannatomczak@interia.pl; 3Department of Clinical Pharmacy and Biopharmacy, Poznan University of Medical Sciences, 60-806 Poznan, Poland; eszalek@ump.edu.pl (E.S.); akarbownik@ump.edu.pl (A.K.)

**Keywords:** AML therapy, microbiome, dysbiosis, drug–microbiome interactions, pharmacomicrobiomics

## Abstract

Increasing evidence demonstrates the mutualistic connection between the microbiome and acute myeloid leukemia (AML) treatment. Drugs disrupt the microbial balance and, conversely, changes in the microbiome influence therapy. A new field, pharmacomicrobiomics, examines the role of the microbiome in pharmacokinetics, pharmacodynamics, and drug toxicity. The multimodal therapeutic management of AML, along with disease-related immunosuppression, infection, and malnutrition, creates the unique microbial profile of AML patients, in which every delicate modification plays a crucial role in pharmacotherapy. While both preclinical and real-world data have confirmed a bilateral connection between standard chemotherapy and the microbiome, the impact of novel targeted therapies and immunotherapy remains unknown. Multi-omics technologies have provided qualitative and mechanistic insights into specific compositional and functional microbial signatures associated with the outcomes of AML therapy, but require a large-scale investigation to draw reliable conclusions. In this review, we outline the role of the microbiome within the therapeutic landscape of AML, focusing on the determinants of post-treatment dysbiosis and its effects on the therapeutic response and toxicity. We explore emerging strategies for microbiota modulation, highlighting their safety and efficacy. Advances in microbiome-based approaches are an inevitable step toward precision medicine in AML. However, clinical research in a well-defined group of immunocompromised patients is needed to study their variable effects on human health and determine safety issues.

## 1. Introduction

Millions of microorganisms inhabiting the human body and their genomes, collectively known as the microbiome, create a functional ecosystem that maintains a mutualistic balance with the host. Cumulative exposures across the lifespan—including diet, medication, environment, and aging—sculpt a unique microbial fingerprint in each individual. Although generally stable, the microbiome can undergo lasting alterations in its taxonomic composition and reduced diversity under specific conditions, potentially generating disease-associated microbial patterns with biomarker potential. Recent advancements in genomics, metagenomics, and metabolomics emphasize the microbiome’s crucial role in immunity, hormones, inflammation, metabolism, intestinal barrier function, and hematopoiesis [1,2]. A microbiome imbalance, referred to as dysbiosis, can influence the onset, progression, and response to the therapy of cancers, including acute myeloid leukemia (AML) [3]. Mechanistically, it not only influences mutagenesis, cell proliferation, and microenvironmental immunomodulation locally [4], but also provides bioactive metabolites that exert systemic health effects [5]. As the understanding of the cancer biology expands, researchers increasingly recognize the polymorphic variability of the microbiome as essential to the core hallmarks of cancer and, therefore, as a promising therapeutic target [6].

Given the significance of the microbiome in health and disease, its effect on the disposition and drug response has garnered significant attention, which is reflected in the increasing number of scientific papers in the field (51,913 publications were reported for search terms “microbiome and therapy” in PubMed.gov as of 30 May 2025). To address this knowledge gap, the emerging field of pharmacomicrobiomics has investigated how the microbiome affects drug pharmacokinetics, pharmacodynamics, and toxicity [7]. Two additional terms further refine this field: toxicomicrobiomics, which examines the microbiome’s impact on drug metabolism and toxicity, and pharmacoecology, which refers to changes in the microbiome following drug administration [8]. In cancer patients, these interactions are especially relevant as they may critically influence treatment outcomes.

AML remains one of the most challenging blood cancers, with a 5-year overall survival (OS) of 30% and a 50% rate of relapse after achieving complete remission (CR) [9]. Originating from somatic genetic and epigenetic alterations in immature myeloid progenitor cells in the bone marrow, it progresses aggressively to create subclonal populations with a high potential for resistance to treatment [10]. While extensive research has elucidated the genetic landscape of AML, we are in the infancy of understanding the microbiome’s place in AML pathogenesis and progression. Compared to healthy controls, treatment-naïve patients with AML exhibit reduced diversity in their gut microbiome [11]. One study noticed significantly higher relative abundances of *Streptococcus* [12]. The analysis of microbiome data in European leukemia populations identified the *Rikenellaceae* RC9 gut group, *Anaerostipes*, *Slackia*, and *Lachnospiraceae* ND3007 group as risk factors for AML [13]. However, understanding their causal relationship with AML biology requires a detailed multi-omics approach.

The pre-existing state of dysbiosis is markedly exacerbated during treatment (Table 1) [14]. Figure 1 highlights the critical points of microbiome disruption and alteration during the clinical course of AML patients.

In addition to leukemia-specific therapies, broad-spectrum antibiotics and supportive medications (analgesics, antipyretics, diuretics, antiemetics, allopurinol, laxatives, antacids, and antidepressants) used in AML markedly disrupt the microbiota [15]. Microbial shifts persist even after hospital discharge and recovery, resulting in new communities highly dissimilar from the baseline [16]. In turn, the microbiome composition affects the outcomes of the treatment. Higher alpha diversity and enrichment in health-associated taxa belonging to the genera *Faecalibacterium*, *Ruminococcus*, *Blautia*, and *Butyricimonas* at diagnosis are predictive for improved the post-induction recovery of platelet, lymphocyte, and neutrophil counts [17]. Dysbiosis has proven importance in complications of intensive therapy, including infections [18,19], by creating an environment conducive to pathogen colonization [20]. Conversely, a favorable microbiota composition has been associated with improved outcomes, reduced treatment-related complications, and enhanced responses to cancer therapies, highlighting its potential as a promising avenue for therapeutic intervention [19,21,22].

Based on previous clinical observations from different cancers, we conclude that the drug’s mode of action or side effects are partly attributable to the composition and function of the microbiome [23]. However, challenges remain in the AML setting. A key obstacle is the lack of a detailed mechanistic understanding of how treatments, especially novel therapies, interact with human biology and the microbial community, limiting further exploration in pharmacomicrobiomics. While the real-world data let us look closer at the microbiome dynamics during the treatment of AML, interventional studies remain largely in preclinical stages, leaving many questions about the safety, implications, and therapeutic relevance of microbiome-centered strategies.

This paper reviews the literature to summarize existing findings on the complex crosstalk between the microbiome and AML therapy. First, we present how AML treatments, particularly chemotherapy and antibiotics, influence the microbiota composition. Then, we focus on the role of the microbiome in modulating AML treatment outcomes, considering underlying mechanisms. Further, we look at prebiotics, probiotics, paraprobiotics, and postbiotics as potential therapeutic strategies. Finally, we outline current and future research directions while addressing the limitations of pharmacomicrobiomics in AML treatment.

## 2. AML Therapy Triggers Dysbiosis

In addition to anticancer therapies, supportive agents—such as antibiotics, analgesics, antiemetics, allopurinol, diuretics, laxatives, proton pump inhibitors, and psychiatric medications—as well as treatment-related gut barrier disruption and immunosuppression, contribute to dysbiosis. Reduced microbial diversity and pathogenic shifts at the phylum, genetic, and metabolic levels characterize this state.

### 2.1. Antileukemic Treatment

Intensive therapy in acute leukemia is prolonged, highly myelosuppressive, and causes significant mucosal barrier damage. The standard induction treatment for AML involves 3 days of daunorubicin and 7 days of cytarabine (3 + 7 regimen). Newer strategies include CPX-351, a dual-drug liposomal encapsulation of these agents, targeted therapies such as *FLT3* inhibitors (midostaurin, quizartinib, and gilteritinib), *IDH* inhibitors (ivosidenib, olutasidenib, and enasidenib), the Hedgehog pathway inhibitor glasdegib, the *BCL-2* inhibitor venetoclax, and the monoclonal antibody gemtuzumab ozogamicin. Consolidation therapy with cytarabine, followed by HSCT in intermediate and adverse-risk patients, constitutes the standard post-remission therapeutic procedure for AML [24]. For patients deemed unfit for intensive therapy, a lower-intensity regimen combining azacitidine and venetoclax remains a viable alternative (Figure 2).

#### 2.1.1. Standard Therapy

The increasing number of studies has documented the negative impact of standard therapy on the microbiota in AML. The post-chemotherapy decline is characterized by an increase in *Lactobacillus*, a decrease in *Blautia*, and the dominance of opportunistic pathogens like *Staphylococcus*, *Enterobacter*, and *Escherichia* in the gut. Long-term analyses of oral and stool microbiomes during intensive chemotherapy show significant intra-patient variability in the microbial diversity, which is linked to a higher risk of infection. This variability is associated with predominant pathogenic genera such as *Staphylococcus*, while more stable states exhibited higher levels of beneficial microbes like *Akkermansia*, *Subdoligranulum*, and *Pseudobutyrivibrio* [19,25]. Microbial disruptions caused by chemotherapy are enduring, with increased levels of *Bacteroides* and decreased levels of *Faecalibacterium* and *Alistipes* lasting up to six months after treatment. The microbiome remains altered from its baseline composition even after antibiotic tapering and post-discharge recovery, failing to return to its original state—suggesting sustained disruption and incomplete restoration [16]. Reduced gut microbiota (GM) diversity persisted despite bone marrow recovery, and subsequent re-induction or salvage therapy further destabilized the ecosystem, leading to compromised colonization resistance, increased susceptibility to *Enterococcus* overgrowth, and a higher risk of infections [26]. Rattanathammethee et al. studied the microbiome in the long term to identify changes during neutropenic fever (NF). *Firmicutes* thrived during NF but declined after bone marrow recovery, contrasting with trends in *Bacteroidetes* and *Proteobacteria*. Compared to pretreatment, *Enterococcus* increased during NF, while *Escherichia* declined. Microbial richness was higher before treatment than during NF and remained at a low level after bone marrow recovery [27]. This study highlights a pathogenic association and suggests the potential predictive value of the GM composition for NF. In contrast to prior findings, Jing Xu et al. reported increased alpha diversity during chemotherapy when comparing the GM and metabolic profiles of AML patients—with and without chemotherapy—to those of healthy controls. The discrepancy with prior findings may be attributed to the absence of antibiotics. The distinct beta diversity profile differentiated three groups. Compared to controls, AML patients displayed an increased *Firmicutes*-to-*Bacteroidetes* ratio, with the significant enrichment of *Collinsella* and *Coriobacteriaceae* as potential biomarkers. Additionally, the *Eubacterium hallii* group was notably enriched in AML patients relative to those undergoing chemotherapy [28].

Medications influence not only the composition of the gut microbiota but also its metabolic activity, thereby affecting the host physiology. Hueso et al. linked the 7 + 3 to gut injury and dysbiosis, evidenced by decreased plasma citrulline (a marker of functional enterocyte mass), short-chain fatty acids (SCFAs), and the fecal bacterial load, except for *Escherichia coli* and *Enterococcus* spp. These features were associated with concurrent histologic impairment in mice with AML [29]. Pötgens et al. have recently highlighted the links between post-treatment gut microbiome changes and cachectic features in AML patients. Elevated systemic inflammation, muscle mass depletion, anorexia, and weight loss coexisted with the transient impairment of the gut barrier function and persistent reduced diversity. *Lactobacillaceae* and *Campylobacter* levels were increased at induction completion, whereas *Enterococcus faecium* and *Staphylococcus* levels were at discharge. Metabolomics analyses indicated reductions in urinary hippurate and fecal bacterial amino acid metabolites [30].

Arsenic trioxide, in combination with all-trans retinoic acid, forms the cornerstone of therapy for acute promyelocytic leukemia. After exposure, patients presented reduced diversity and notably decreased *Bifidobacterium adolescentis* and *Lactobacillus mucosae*. Additionally, arsenic trioxide prompted the development of resistance genes in *Bacteroides fragilis*, a prevalent gut bacterium [31].

#### 2.1.2. Novel Therapies

Limited research has considered the microbiome in the context of novel therapies. Recently, researchers have paid close attention to CPX-351, which positively impacts the mucosal barrier, microbiome composition, and immune homeostasis in an animal AML model. The effects were mediated via the activation of the aryl hydrocarbon receptor-IL-22-IL-10 pathway and the production of immunomodulatory metabolites by anaerobes, which improve gut barrier integrity, decrease local inflammation, and regulate the local microbial community [32]. Liu W et al. assessed GM and metabolite profiles of serum and urine in 29 patients with AML before, during, and post-therapy in the following groups: chemotherapy (cytarabine + idarubicin) +/−venetoclax; azacytidine + venetoclax/gilteritinib/afatinib; and venetoclax + gilteritinib. Treatment reduced GM diversity, which did not recover until the next cycle. *Enterococcus* expansion was associated with a high risk of NF, whereas decreases in *Anaerococcus* and *Dialister* and an increase in *Enterobacteriaceae* predicted CR. The authors did not compare specific therapies [33]. A taxonomic comparative analysis in patients with colorectal, stomach, breast, lung, melanoma, and lymphoid neoplasms and AML did not identify any bacteria that differentiated patients tested before and after immunotherapy or chemotherapy in any of the groups, except for breast cancer. The patients with AML received different treatments (cytarabine + idarubicin, cytarabine + daunorubicin, or cytarabine + gemtuzumab ozogamicin), either with or without autoHSCT [34]. At ASH 2024, Italian researchers presented findings from a prospective analysis of GM and blood and fecal metabolomics in myeloid neoplasms (AML n = 198), at diagnosis and follow-ups. In treated patients, both the microbiome composition and metabolomic profiles shifted significantly, alongside gut mucosal injury from chemotherapy. Compared to the 3 + 7, hypomethylating agents +/−venetoclax better preserved the microbial balance, correlating with a lower risk of infections [35].

### 2.2. Allo-HSCT and Microbiome

Allo-HSCT includes a conditioning regimen, typically involving chemotherapy and radiotherapy, followed by the infusion of donor hematopoietic cells. This procedure significantly alters the composition and function of the microbiome, influencing clinical outcomes.

#### 2.2.1. Conditioning

The impact on the intestinal microbiome depends on the specific conditioning regimen. Montassier et al. reported a post-conditioning decline in α-diversity, characterized by a significant decrease in *Firmicutes* and an increase in *Escherichia* spp., correlated with the conditioning intensity [22,36]. High-intensity myeloablative regimens, based on total body irradiation (TBI) or busulphan, showed the highest depletion of commensal bacteria and expansion of *Enterococcus*, while cyclophosphamide/fludarabine/TBI200 preserved pre-transplant microbial compositions most effectively [37]. Similarly, the myeloablative group showed the most profound decline in the alpha diversity, species, gene, and metabolic richness following allo-HSCT [38].

Radiation acts as a stressor to the gastrointestinal tract’s microbial ecosystem, potentially leading to mucositis, diarrhea, and fatigue in cancer patients. The systematic review of studies on ionizing radiation in animals revealed an increased abundance of *Lactobacillaceae* and *Staphylococcaceae* families, alongside decreased levels of *Lachnospiraceae*, *Ruminococcaceae*, and *Clostridiaceae* [39]. A murine model examining total body irradiation at 0, 4, 8, and 12 Gy revealed dose-dependent changes in intestinal tissues and GM. The abundance of *Proteobacteria*, *Escherichia-Shigella*, *Eubacterium xylanophilum* group, and *Lactobacillus murinus* correlated with the radiation dose. Dysbiosis persisted through recovery, but may be alleviated by a 14-day administration of probiotics [40]. In mice, radiotherapy and melphalan therapy increased the levels of bacteria with mucin-degrading capabilities, particularly *Akkermansia* spp., and fever risk, similar to cytotoxic chemotherapy in HSCT neutropenic recipients, even without antibiotics [41].

GM may enhance the radiosensitivity and strengthen anticancer responses; in contrast, broad-spectrum antibiotics reduce radiotherapy efficacy. Proposed mechanisms include direct and SCFA-mediated anti-tumor immunity, as well as the modulation of the hypoxic tumor microenvironment. The GM contributes also to radiation-induced toxicities, including diarrhea, mucositis, and pulmonary complications, highlighting its potential as a therapeutic target [42].

Rashidi et al. compared GM changes between patients with intensively treated acute leukemia and recipients of allo-HSCT. Microbial diversity decreased and *Enterococcus* increased in both cohorts, while *Lactobacillus* rose only in the leukemia group [43], emphasizing the need for a personalized approach to address distinct microbiota changes in individual patients.

#### 2.2.2. T-Cell Depletion

Alloreactive T lymphocytes in the graft worsen dysbiosis by damaging intestinal epithelial cells. T-cell depletion can mitigate this effect. The decline in the diversity of the intestinal microbiota during allo-HSCT remains generally consistent across various graft-versus-host disease (GVHD) prophylaxis regimens, including post-cyclophosphamide-based and anti-thymocyte globulin-based approaches [44,45]. However, variability in the transplant indications, comorbidities, antimicrobial exposure, and conditioning intensity hinders the direct comparison of GVHD prophylaxis effects across clinical studies.

### 2.3. Antibiotics

Due to the high mortality related to infections, an appropriate antibiotic policy is a key element of AML treatment. However, the overuse or misuse of antibiotics leads to intestinal dysbiosis, which is associated with recurrent *Clostridioides difficile* infections (CDI), systemic infections, and worse clinical outcomes. Table 2 summarizes the clinical studies investigating the effect of antibiotics on the microbiome in AML.

Antibiotic-induced dysbiosis has been linked to treatment outcomes and complications in patients with AML with the strongest body of evidence relating to gastrointestinal GVHD. The use of broad-spectrum antibiotics such as imipenem–cilastatin and piperacillin–tazobactam for NF was associated with increased GVHD-related mortality in clinical trials and murine models [48]. Antibiotic prophylaxis and therapy promoted enterococcal dominance, correlating with a higher risk of gastrointestinal GVHD [47]. In addition to compositional changes, HSCT patients’ microbiomes showed an accumulation of antimicrobial resistance (AMR) genes, closely associated with aGVHD. Interestingly, AMR gene patterns could not be fully explained by antibiotic use alone, suggesting more complex mechanisms of resistance acquisition [53].

Of particular concern is the increased risk of infection associated with antibiotics. Enterococcal domination, exacerbated by metronidazole use, was linked to a higher incidence of vancomycin-resistant *Enterococcus* bacteremia [46]. Prolonged carbapenem use (>72 h) significantly reduced gut microbial alpha diversity at the time of neutrophil recovery and was associated with a higher risk of infection within the subsequent 90 days [54]. In contrast, fluoroquinolone prophylaxis was independently associated with a 55% reduction in Gram-negative BSIs, likely due to the suppression of intestinal colonization by *Gammaproteobacteria* [55]. Rifaximin has emerged as a potentially beneficial alternative, associated with lower enterococcal prevalence, a reduced incidence of gastrointestinal GVHD, lower transplant-related mortality, and improved OS, compared to regimens involving ciprofloxacin plus metronidazole, piperacillin–tazobactam, meropenem with vancomycin, ceftazidime, or multiple systemic antibiotics [49].

The association between specific antibiotics and AML treatment outcomes remains difficult to establish due to numerous confounding variables. By inducing dysbiosis, antibiotics may disrupt drug metabolism and impair immune function, potentially compromising the therapeutic efficacy. However, a comprehensive assessment of treatment effects must consider both the indications for and the specific use patterns of antimicrobial agents.

Prophylactic and empirical antibiotic practices for AML vary globally. Fluoroquinolones prophylaxis was recommended in 2005 by the European Conference on Infections in Leukemia (ECIL) for high-risk patients, such as those with AML or undergoing HSCT [56]. However, its common use has increased resistance to Gram-positive, Gram-negative, and CDI [57]. Given the absence of an overall mortality benefit, its clinical value remains increasingly contested [58]. Monotherapy with rifaximina offers a safer alternative to current non-selective prophylaxis in AML, preserving intestinal microbiome diversity even when used alongside broad-spectrum antibiotics [49]. It acts eubiotically by reducing bacterial virulence, inhibiting adherence to epithelial cells, and decreasing mucosal inflammation [59]. Its localized gastrointestinal activity targets aerobic Gram-negative pathogens while sparing the anaerobic microbes, representing over 99% of the gut microbiome. In clinical settings, rifaximin minimizes the disruption of GM, promoting beneficial bacteria like *Bifidobacteria* and *Lactobacilli* [59].

The current ECIL-10 recommendations for empirical therapy in AML patients include escalation (anti-pseudomonal cephalosporins, piperacillin–tazobactam, cefoperazone–sulbactam, or piperacillin plus gentamicin) and de-escalation strategies (carbapenem monotherapy, anti-pseudomonal β-lactams with aminoglycosides or quinolones, colistin with β-lactams ± rifampicin, glycopeptides, or newer agents for resistant Gram-positive infections) [60]. According to the increasing incidence of extended-spectrum beta-lactamase-producing infections, carbapenems with broad-spectrum activity against Gram-positive and Gram-negative organisms are commonly selected. However, they also pose a significant danger by reducing beneficial gut commensals, especially those that uphold the mucus layer, and contribute to the rise of antibiotic resistance [61].

*Enterococcus* species are common commensals of the gastrointestinal tract. Due to their intrinsic resistance to multiple antimicrobial classes, including cephalosporins, lincosamides, fluoroquinolones, aminoglycosides, clindamycin, and trimethoprim/sulfamethoxazole, *Enterococcus* has become the most common dominating genus in AML [46,62]. Exposure to cephalosporins, fluoroquinolones, and vancomycin is a risk factor for colonization with vancomycin-resistant *Enterococcus* [63,64]. Patients with *Enterococcus* domination tended toward *E. faecalis* and *E. faecium* bacteremia, gut GVHD, reduced OS, and increased treatment-related and relapse-related mortality [62]. Alongside fluoroquinolone-resistant *Enterobacterales*, extended-spectrum beta-lactamase producers, and multidrug-resistant *Pseudomonas aeruginosa*, these pathogens now contribute to higher mortality rates in AML patients.

Clinician adherence to recommendations against routine bacterial prophylaxis and early de-escalation remains suboptimal. Improving antibiotic regimens to minimize harm to the GM while maintaining the therapeutic efficacy presents a challenge for future research endeavors.

## 3. Impact of the Microbiome on the Anticancer Treatment in AML

Interindividual variability in the drug response and toxicity poses a significant challenge in cancer therapy. To address this, pharmacomicrobiomics, which explores the relationship between variations in the human microbiome and pharmacological responses, has rapidly developed in recent years, revealing numerous underlying mechanisms. The microbiome directly influences a drug’s availability and activity and the sensitivity of the host to related toxic effects [65]. Microbiome-derived metabolism refers to the biochemical transformation of xenobiotics by microbiome-produced enzymes: the activation of a prodrug or conversion to inactive or toxic metabolites. Additionally, the GM may influence drug metabolism by competing with drug molecules for metabolizing enzymes, altering the levels of the host’s drug-metabolizing enzymes in the liver and intestine, or producing enzyme-inducing metabolites derived from the diet [66]. Indirectly, the microbiome improves therapeutic outcomes by modulating the host immune system, as observed in enhanced antigen presentation and better T cell function during a PD-1/PD-L 1 blockade in immunotherapy. Responders showed higher gut microbial diversity and increased levels of *Clostridiales*, *Ruminococcaceae*, and *Faecalibacterium* [67]. Understanding various mechanisms of action in pharmacomicrobiomics has led to the idea of targeting the microbiome to predict responses, boost efficacy, and reduce side effects of cancer treatments.

Microbiota chemically alter the metabolism of several drugs. In an analysis by Zimmermann et al. testing the metabolic capacity of 76 gut microbial strains, 176 of 271 orally administered drugs (66%) were metabolized by at least one bacterial strain [68]. Some of these drugs are used in hematology. Variations in the microbiome phenotype influence the efficacy and toxicity profiles of drugs such as cyclophosphamide, methotrexate, platinum compounds, cisplatin, doxorubicin, cladribine, rituximab, and anti-CD19 CAR T cell therapy [21,69,70,71,72,73,74]. This is also supported by the reduced anticancer immune response related to dexamethasone and corticosteroid metabolism, which results from the adverse effects of antibiotics on the microbiome [68]. In vivo data show that *E. coli* can inhibit chemotherapeutics like cladribine, vidarabine, doxorubicin, idarubicin, daunorubicin, etoposide phosphate, and mitoxantrone, while enhancing the effects of drugs such as mercaptopurine, fludarabine phosphate, and 5- 5-fluorocytosine [72]. Bacteria can also influence cancer cell characteristics and modify chemotherapy sensitivity through toxin production. In AML patients, *Staphylococcus aureus*-derived enterotoxins A and B increase AML cell proliferation, migration, invasion, and resistance to cytarabine. By dysregulating immune-related genes, these toxins may help AML cells evade hostile environments, potentially via the endoplasmic reticulum stress signaling pathway [75]. Only a few clinical trials have focused on the microbiome’s impact on the drug response in AML patients (NCT05596968, NCT04214249).

## 4. Impacts of the Microbiome on Treatment Complications in AML

The findings from clinical studies in acute leukemia point to a role of the microbiota in shaping chemotherapy complications such as organ toxicity, neutropenia, infections, GVHD, and gastrointestinal dysfunction. Emerging evidence identifies alpha diversity and GM variability as potential predictors of infection risk during and after chemotherapy [19,25,54]. Pre-treatment *Proteobacteria* abundance correlates with NF [76] and low diversity with a high *Enterococcus* spp. level relates to increased systemic inflammation and intestinal epithelial integrity injury [77]. The intricate relationship between a pre-existing microbiome composition and treatment-related complications influences the overall outcomes after allo-HSCT in leukemia. For instance, increased bacterial diversity [22] and a higher relative abundance of *Blautia* were correlated with reduced mortality [78], whereas lower gut microbial diversity was associated with decreased OS, a higher risk of treatment-related mortality, and an increased risk of GVHD-related mortality [79]. Besides lower microbial diversity, a compositional shift characterized by a decline in beneficial bacteria (*A. muciniphila*, *Blautia*, and anti-inflammatory *Clostridia* [80]), as well as an increase in *Enterococcus* [81], was predictive of aGVHD in another study. Also, microbiota-derived metabolites, including SCFAs, tryptophan and its derivatives, choline metabolites, tyrosine, and bile acids, affected the severity and prognosis of aGVHD [82]. Finally, a higher abundance of *Eubacterium limosum* was linked to a reduced risk of relapse and disease progression in patients undergoing allo-HSCT [83].

Figure 3 provides a comprehensive summary of the relationships between microbiome changes, AML treatment, and clinical outcomes, highlighting potential underlying mechanisms.

## 5. The Therapeutic Potential of the Microbiome

Based on evidence outlining the role of the microbiota in mediating the effects of drugs [84,85,86], researchers have intensified efforts to develop strategies for manipulating the microbiome for preventive and therapeutic purposes. However, in high-risk patients, such as those with leukemia, numerous practical and ethical challenges have hindered clinical implementation.

### 5.1. Antibiotic Stewardship

Antimicrobial stewardship—coordinated strategies to optimize antimicrobial use, improve outcomes, reduce resistance, and limit the spread of MDRO—is essential to the comprehensive care of leukemia patients. This section discusses emerging strategies that aim to balance effective infection control with microbiome preservation, ultimately enhancing treatment outcomes.

Broad-spectrum antibiotics (vancomycin, imipenem, colistin, ampicillin, beta-lactams, quinolones, sulfonamides, streptomycin, and neomycin) have been linked to adverse effects on cancer therapy, possibly through microbiome-mediated mechanisms. Therefore, using narrow-spectrum antibiotics to selectively target harmful bacterial species, known as “targeted microbiome-sparing antibiotics,” could play an adjunctive role in modulating the therapy response.

A lack of mortality benefit and rising resistance with fluoroquinolone prophylaxis support revising current strategies, with rifaximin emerging as a promising alternative [87,88,89].

Empirical antibiotic strategies are often tailored to general institutional antimicrobial sensitivity patterns while overlooking adjustments based on an individual patient’s microbiological colonization profile. Characterizing microbiome structures upfront can help guide personalized prophylaxis and treatment.

Following current guidelines, the early de-escalation and discontinuation of antibiotics for NF are effective without worsening outcomes in hematology wards [90]. Limitations in carbapenem use resulted in decreased vancomycin-resistant *E. faecium* colonization, BSI, hospitalization duration, and costs in the patients with hematological malignancies unit [91]. A study in AML and MDS patients undergoing induction chemotherapy confirmed the safety of the withdrawal of empirical antibiotics after 72 h in hemodynamically stable patients [92].

Further research is needed to evaluate the effectiveness of new antibiotic compounds and microbiome-preserving strategies, such as beta-lactamase enzymes that act locally within the gastrointestinal tract, and non-specific adsorbents to sequester the antibiotics within the colon [93,94].

### 5.2. Diet

Diet is one of the most significant factors influencing the human microbiome. Microbiome-mediated mechanisms partially explain the dietary impact on the drug response and side effects, as seen in immune [95] and radiation-induced anti-tumor responses [96] and chemotherapy-induced mucositis [97,98]. Targeted dietary interventions may return dysbiosis to a healthy state or tend to change in favor of the oncologic patient.

In recent years, various diets, including fasting or fasting-mimicking diets, ketogenic diets, and fiber-rich Mediterranean diets, have been studied to enhance the effectiveness of cancer treatments while reducing their toxicity. Research has shown that fasting interventions are linked to increased levels of beneficial gut bacteria such as *Faecalibacterium*, *Roseburia*, *Butyricoccus*, and *Coprococcus*. These bacteria are significant producers of SCFAs. *F. prausnitzii* has been shown to improve chemotherapy efficacy in preclinical studies and clinical trials involving cancer patients (melanoma [67], thyroid [99], and colorectal [100,101]). *Akkermansia*, *Roseburia*, and *Ruminococcaceae* were all elevated in patients on a ketogenic diet [102]. Preliminary findings from a study investigating the administration of a high-fat ketogenic diet before and during induction chemotherapy in patients with AML demonstrated favorable tolerability and an increase in DNA damage within leukemic blasts. Additionally, the diet exhibited protective effects against senescence in healthy lymphocytes. However, the study did not evaluate its impact in the microbiome context [103]. A fiber-rich diet improved the efficacy of immunotherapy [104] and reduced the toxicity of 5-fluorouracil treatment [98] by prompting the gastrointestinal microbiome to produce SCFAs. In a mouse model, the Mediterranean diet, compared to the Western diet, was associated with greater microbiome diversity and higher abundances of *Clostridium*, *Lactobacillus*, *Oscillospira*, and *Faecalibacterium*, while showing lower levels of *Coprococcus* and *Ruminococcus* [105]. These results were consistent with human studies that demonstrated an increased abundance of *Lachnoclostridium*, *Enterorhabdus*, and *Parabacteroides*, along with enhanced SCFA production [106]. A study involving 41 pediatric HSCT patients found that a higher pre-treatment intake of soluble fiber, iron, breast milk, bazlama, yogurt, onion, parsley, and bulgur was associated with earlier neutrophil engraftment, a lower incidence of NF episodes, and shorter durations of total parenteral nutrition [107]. Despite promising findings, adhering to dietary interventions can prove challenging for malnourished AML patients undergoing aggressive treatments. Therefore, maintaining a balanced diet with an adequate nutrient intake is essential for recovery and for alleviating treatment-related complications. The low-bacterial diet, designed to minimize infection risk, has gained popularity, yet it has faced criticism. A review of 12 studies on low-bacterial diets found no significant benefits in reducing infection or mortality rates. Additionally, this diet often resulted in a diminished quality of life due to less appealing food options and limited variety, which negatively affected the overall nutritional status [108]. This diet may restrict key dietary components, particularly fiber, leading to dysbiosis and decreased SCFA production. Given the significant role of nutrition in AML patients, the current U.S.A Food and Drug Administration (FDA) guidelines on nutrition highlight the importance of maintaining food safety through proper handling rather than dietary eliminations [109]. More specific recommendations that incorporate microbiome-related changes and their impact on treatment on AML warrant further research.

### 5.3. Microbiota-Targeted Nutritional Approaches

Microbiota-modulating dietary modalities, including prebiotics, probiotics, paraprobiotics, and probiotics, have achieved favorable results in non-hematological patients (lung, genitourinary, colorectal, renal, prostate, melanoma, solid, gastrointestinal, pancreatic, hepatocellular, head and neck, and breast cancers [110]). In patients with AML, the effects and safety of these interventions are complicated by profound immunological disturbances, limiting research primarily to in vitro and animal studies, with only a few involving humans.

#### 5.3.1. Probiotics and Prebiotics

Certain probiotic species may promote apoptosis in myeloid leukemia cells. *Lactobacillus reuteri* suppresses NF-κB-dependent proliferation, reduces survival signaling, and enhances MAPK pathway activity, thereby promoting cell death [111]. *L. casei rhamnosus* killed the cells of the human monocytic leukemia line, THP-1 [112]. In vitro, *Lactobacillus kefiri* P-IF-based kefir induced apoptosis in multidrug-resistant human myeloid leukemia cells [113]. Curcumin enhanced sensitivity to cytarabine and influenced the intestinal microbiota without directly affecting the AML cell lines in a mice model. A metagenomic analysis indicates an increase in beneficial bacteria, such as *Lactobacillus acidophilus*, *Bifidobacterium bifidum*, and *Lactobacillus reuteri*, alongside a decrease in pathogenic species like *E. coli*, *A. muciniphila*, and *Bacteroides fragilis*. Additional mechanisms may include the enhancement of the intestinal integrity and a reduction in bacterial translocation into the bloodstream [114]. Combining both probiotic and prebiotic properties, kimchi inhibited HL-60 (human acute promyelocytic leukemia) cell proliferation by inducing apoptosis and disrupting the mitochondrial membrane potential, suggesting anticancer potential [115]. Concerns about systemic infections, pathogenicity, excessive immune stimulation, toxicity, metabolic activity, and horizontal gene transfer continue to restrict the medical use of probiotic products in vulnerable patients. For instance, a study found that oral treatment with *Lactobacillus* species in individuals with compromised immune systems led to bacteremia [116].

#### 5.3.2. Paraprobiotics

Researchers are exploring paraprobiotics—inactivated bacterial cells or extracts—as safer alternatives to live probiotics, with some already available commercially. Their various benefits—including the absence of risk for bacterial translocation, prevention of transferring antibiotic resistance genes, and the convenience of standardization, production, transportation, and storage—render them a viable alternative for patients with weakened immune systems [117].

#### 5.3.3. Postbiotics

Postbiotics cover a group of different chemical compounds, including vitamins, organic acids, SCFAs, and amino acids. Their anticancer effects include selective cytotoxicity against tumor cells, antiproliferative action, apoptosis induction, antioxidant activity, gut microenvironment modulation, inflammation reduction, intestinal barrier protection, and immune response regulation [118]. SCFAs play a critical role in numerous biological processes, with acetic acid, butyric acid, and propionic acid constituting the three primary types. Among these, butyrate functions as a histone deacetylase inhibitor, effectively normalizing epigenetic imbalances. Furthermore, butyrate has been shown to modulate genes associated with apoptosis and the cell cycle, impede the proliferation of tumor cells, and counteract resistance linked to anti-apoptotic regulators, such as MCL-1 [119], a known mechanism of resistance to venetoclax [120]. The anti-cancer potential of sodium butyrate has driven trials on its efficacy in AML treatment. Pulliam et al. described the apoptotic effect of butyrate in human acute leukemia cells by the activation of caspase-3, reduction in cell viability, and lowering the concentration of the chemokines CCL2 and CCL5 [121]. Its implementation enhanced the efficacy of venetoclax [122]. Administering sodium butyrate with TRAIL enhanced the killing effect on t (8;21) AML cells [123]. Clinical trials are needed to explore the efficiency of this approach in AML patients.

#### 5.3.4. Modulating Chemotherapy Toxicity with Pro-, Pre-, Para-, and Postbiotics

With limited options for managing chemotherapy toxicity, microbiome modulation offers a promising therapeutic strategy. In children with acute leukemia, daily supplementation with probiotic *Lactobacillus rhamnosus* reduced abdominal distension, intestinal constipation, and nausea during chemotherapy [124]. The *Bifidobacterium breve* strain Yakult diminished the incidence of fever in pediatric leukemic patients [125]. Enteral supplementation enriched with glutamine, fiber, and oligosaccharides resulted in fewer days of diarrhea and mucositis of grade 3–4 in HSCT patients [126]. The administration of resistant starch and the commercially available prebiotic mixture GFO to allo-HSCT recipients, starting from pretransplant conditioning through day 28 post-transplant, effectively mitigated mucosal injury and reduced the incidence of diarrhea. This intervention positively impacted gut microbiome diversity, increased the population of butyrate-producing bacteria, and elevated fecal butyrate concentrations after the transplant [127]. In a Phase II study involving AML patients undergoing high-dose chemotherapy and radiation followed by HSCT, the use of *Lactobacillus brevis* CD2 for prophylaxis was both safe and effective. Of the 31 patients enrolled, only six developed severe oral mucositis [128]. In a recent retrospective analysis, viable *Bifidobacterium* tablets significantly reduced the incidence and duration of grades 1–2 oral mucositis during the transplant process without impacting HSCT outcomes [129]. In contrast, the first randomized trial of HSCT patients supplemented with *Lactobacillus rhamnosus* GG found no significant changes in the GM or GVHD incidence [130].

Though currently classified as health products, microbiome-based interventions may soon achieve a drug status, with defined indications, dosages, and safety profiles as research advances.

### 5.4. The Role of the Microbiota in GVHD: From Mechanisms to Therapies

GVHD remains a major cause of morbidity and mortality following HSCT. Its pathogenesis primarily involves the presentation of host alloantigens to naïve donor T cells, which then mount an immune response characterized by progressive tissue injury. The gastrointestinal tract plays a central role as an early site of alloreactivity. Various antigen-presenting cells—such as dendritic cells, mesenchymal stromal cells, and intestinal epithelial cells—actively participate in initiating and sustaining pathogenic T cell activation and differentiation [131]. However, many of these factors may modulate GVHD through microbiome-related mechanisms, particularly within the gastrointestinal tract, the most densely populated and diverse microbial ecosystem in the human body. The dynamic interplay between treatment regimens, gut integrity, and microbiota composition forms a tightly interconnected triad that critically influences immune homeostasis. The disruption of the intestinal epithelial barrier—often resulting from chemotherapy, radiotherapy, or antibiotic exposure—facilitates the translocation of pathogen-associated molecular patterns from the microbiota, as well as damage-associated molecular patterns from injured host tissues. This influx of microbial and host-derived signals amplifies local inflammation, creating a pro-inflammatory milieu that promotes the onset and progression of GVHD [132].

Beyond epithelial barrier disruption, dysbiosis itself is a significant driver of GVHD. Both preclinical and clinical studies have proposed multiple microbiome-related mechanisms underlying the microbiome T (8;21) AML cells GVHD association: (1) the modulation of MHC class II expression [133], (2) immunological tolerance control [134], (3) immunoregulation (production of IL-17 and IL-22 by Th17 cells in the lamina propria; Th17 and Treg cells balance) [135,136], (4) the activation of Toll-like receptors [137], (5) the regulation of cytokine profiles (activation of inflammasome, production of IL-18, Th1/Th2 balance) [138], and (6) microbial metabolites—mediated mechanisms (SCFAs modulate immune responses by inhibiting NF-κB signaling, enhancing IL-10 expression, and promoting regulatory T cell differentiation) [139,140,141].

Numerous preclinical studies and clinical investigations have explored the relationship between the microbiota composition and the risk and severity of GVHD. Systematic reviews and meta-analyses in both pediatric [142] and adult populations [143] have identified certain characteristic microbial signatures associated with GVHD-related dysbiosis. The most consistent findings include reduced microbial diversity and an increased dominance of genera such as *Lactobacillales*, *Staphylococcaceae*, *Enterobacteriales*, and *Enterococcus*. These changes often coincide with a depletion of bacterial taxa linked to butyrate synthesis, notably *Clostridia*, *Lachnospiraceae*, *Blautia*, *Bacteroides*, and *Akkermansia muciniphila*.

Although pinpointing a single causative microorganism remains challenging, some studies have attempted to define stereotyped pathogenic microbial consortia. Nevertheless, these microbial patterns have not yet been consistently replicated across independent research groups, highlighting the complexity and heterogeneity of GVHD-related dysbiosis.

Growing scientific and clinical evidence supports microbiota-targeted strategies to mitigate GVHD. Therapeutic approaches focus on microbial metabolites such as SCFAs and 3-indoxyl sulfate [144], antibiotic use optimization (e.g., narrow-spectrum agents, a reduced duration, and novel β-lactams), and microbiome-supportive diet interventions including prebiotics, probiotics, and symbiotics [126,145]. Fecal microbiota transplantation (FMT) has a growing interest among hematologists as a promising therapeutic option for GVHD; however, further research is needed to optimize its application and confirm its safety and efficacy in immunocompromised patients [146].

### 5.5. FMT as Full Microbiome Restoration: A Targeted Strategy Against Clostridioides Difficile Infection

CDI poses a significant clinical challenge in patients with hematologic malignancies. Among patients with AML, reported CDI incidences range from 4.8% to 9%, while in recipients of allo-HSCT, the rate rises markedly to 14–30.4%. In conjunction with the underlying disease burden, CDI exacerbates morbidity and contributes to treatment-related mortality, which may approach 20% in this population [147].

Microbiome and CDI are closely related. Balanced GM protect against colonization with pathogens by competing for nutrients, providing inhibitory substances, and regulating the host’s immune response. An imbalance between beneficial and harmful bacteria promotes the transformation of *C. difficile* spores into vegetative forms. First of them are characteristic for a resistance to antibiotics and transmission, whereas vegetative cells produce toxins (toxin A, B, and binary toxin), causing disease. 

While numerous studies have addressed this topic broadly, data specifically focused on patients with AML remain limited. In this population, several factors—including the use of proton pump inhibitors, immunosuppressive agents, impaired mucosal immunity, and the disruption of the intestinal epithelial barrier—may significantly increase CDI susceptibility. However, the most critical contributor remains the widespread use of broad-spectrum antibiotics, which profoundly disturb the gut microbial diversity, density, and metabolic function, thereby promoting *C. difficile* colonization and overgrowth. While nearly all antibiotics, including those used to treat CDI (e.g., metronidazole and vancomycin), can elevate the infection risk, cephalosporins, clindamycin, ampicillin, amoxicillin, penicillin, and fluoroquinolones are associated with the most substantial alterations in the GM [148]. Newly established ecosystem in the gut creates an environment conducive to *C. difficile* colonization and proliferation [149]. Conversely, recurrent *C. difficile* infections further exacerbate gut dysbiosis, perpetuating a harmful cycle [150]. FMT aims to restore the balance of the GM, which may facilitate the clearance of multidrug-resistant organisms, including extended-spectrum beta-lactamase-producing *Escherichia coli*, vancomycin-resistant *Enterococcus*, and carbapenem-resistant *Enterobacteriaceae* [151]. Currently, recurrent *C. difficile* infection remains the only formally approved indication for FMT in clinical guidelines for both adults and children. Nevertheless, guidelines for managing *C. difficile* infection in hematology populations advise against FMT, based on the absence of randomized controlled trials in immunocompromised patients and challenges related to administration [152]. Despite these concerns, emerging clinical evidence supports the safety and feasibility of FMT in selected cases, underscoring the need for broader validation in future trials [153,154].

Completed and ongoing clinical trials on the microbiome in AML, including microbiome-targeted interventions, are listed in Table 3.

## 6. Challenges, Research Directions, and Clinical Implications

As a result of the progress made over recent years, microbiome-focused approaches have become a game-changer for medicine. Following the leading trends in cancer therapy, integrating the microbiome in multimodal management is inevitable. Despite an improvement in the treatment outcomes, AML remains a disease with the highest mortality rates, and further advancements appear to have plateaued with the current available methods. In line with molecularly targeted therapies, individual microbiome-tailored interventions may support the transition toward personalized medicine by both optimizing the impact of conventional therapies and mitigating off-target effects. Building on both foundational knowledge and current findings, we are the first to comprehensively position the microbiome within the therapeutic landscape of AML from multiple perspectives. Due to the relatively limited and inconsistent data, a comprehensive summary is crucial for guiding the direction of future research.

Collectively, evidence from both basic science and clinical studies demonstrates a bidirectional relationship between cancer treatment and patients’ microbial profiles. Despite differences in methodologies, treatment protocols, and clinical factors that can bias the analysis, some findings remain consistent. These typically indicate a reduction in commensal flora, the dominance of a single species, and decreased intestinal microbiota diversity after treatment. The high complexity of the microbiome’s compositional and functional structure in leukemic patients prevents the unequivocal identification of a beneficial profile, but SCFA producers are generally linked to positive results. In contrast, the expansion of *Enterococcus* is commonly associated with bloodstream infections, GVHD, and mortality. Considering the progress in understanding complex microbiome–host interactions, we recognize that we are currently witnessing only the beginning of a boom fueled by breakthroughs in cancer treatment. The number of uncertainties highlights the growing need for extensive and long-term research to draw reliable conclusions. In this section, we explore the primary gaps and unknown frontiers of microbiome research (Figure 4).

Microbiome-focused studies face several methodological limitations. Most have evaluated participants at various stages of treatment (before, during, after induction, after HSCT, etc.) without clearly distinguishing between different regimens or drugs. Studies often fail to account for confounding factors such as diet and antibiotics. Since animal models only partially reflect the human microbiome ecosystem, clinical trials are essential to establish the true link between the microbiome and AML therapies and to identify microbiome-specific targets for treatment.

Currently, the role of microorganisms in cancer therapy goes far beyond microbiome issues. While the synergistic effect of combining traditional therapies with tumor-targeting or immunotherapy has significantly improved outcomes in oncology, challenges such as limited clinical response rates, resistance, and off-target effects of drugs continue to hinder the optimal clinical efficiency. Thanks to their unique properties, bacteria play an active role in modern strategies to overcome these drawbacks [154]. The therapeutic potential of microorganisms includes the following: (1) immunotherapeutic applications utilizing live, attenuated, or genetically modified bacteria, either alone or with conventional treatments; (2) the targeted delivery of anticancer agents; (3) the bacterial expression of tumor-specific antigens; (4) the delivery or expression of suppressor genes, anti-angiogenic genes, and suicide genes; (5) RNA interference mechanisms; and (6) the activation of pro-drugs through bacterial cleavage [155]. Both preclinical and clinical trials with novel bacterial therapeutics primarily focus on solid tumors (bladder, prostate, melanoma, lung, gastric, colon, renal, breast, and cervical cancers) with an achievable tumor microenvironment [156]. Microbial therapy for hematologic neoplasms is currently in its early stages. The difficulty in implementing bacterial-based therapies for hematopoietic neoplasms stems from the distribution of cancer cells in the bone marrow and blood circulation, making systemic chemotherapy the most relevant option. Among the few therapies of proven significance, L-asparaginase, an enzyme produced by bacteria such as *Escherichia coli*, *Bacillus subtilis*, *Streptomyces*, or *Erwinia species*, demonstrates efficacy in treating acute lymphoblastic leukemia and lymphosarcoma [157]. Meirong Li et al. reported that an attenuated *Salmonella typhimurium* strain, VNP20009, can induce apoptosis in leukemia cells both in vivo and in vitro, inhibit the proliferation of MLL-AF9-induced AML cells, and prolong the survival of AML-carrying mice [158]. Given the complex immune mechanisms involved in leukemia, advancing bacteria-based immunotherapy remains a promising research area. The leukemia-permissive microenvironment of the bone marrow, where pathological leukemic stem cells reprogram mesenchymal stem cells to support leukemia progression, chemoresistance, and relapse, presents a potential therapeutic target in AML.

Not only bacteria, but also other microbes such as archaea, fungi, and viruses inhabit the human body. While many efforts have been made to study bacteria, the issue of non-bacterial microbes in health and disease is still unknown, representing the ‘dark matter’ of the microbial ecosystem. Currently, the role of these non-bacterial microbes in cancer remains elusive, not to mention depicting their role in therapy. To fully understand the host–microbiome complexity, future research should explore the entire microbiome as an integrated ecosystem, focusing on both community structure and functional traits, and emphasizing interspecies interactions. Finally, comprehensive sampling and sequencing, both spatially and temporally, will complete the changing picture of the microbial ecosystem throughout treatment in AML patients.

In addition to the best-studied intestinal toxicity, the human microbiota may be associated with other chemotherapy-induced side effects. The study by Cuozzo et al. demonstrated the effectiveness of the probiotic formulation, SLAB51, in preventing paclitaxel-induced neuropathy by enhancing the expression of opioid and cannabinoid receptors in the spinal cord, reducing nerve fiber damage in the paws and regulating the concentration of proinflammatory cytokines in the serum [159]. A randomized trial involving 159 breast cancer patients yielded compelling evidence that probiotic supplementation reduced the incidence of chemotherapy-related cognitive impairment and enhanced cognitive function, potentially through the modulation of plasma metabolites [160]. Finally, animal model studies have linked the gut microbiome with therapy-induced inflammatory and neuropathic pain [161]. Very little is known about the efficiency of microbiome-mediated intervention in managing complications in AML patients. The horrendous gap in knowledge refers to some specific drug-related side effects, such as well-established anthracycline-induced cardiotoxicity, novel issues of cardiovascular adverse effects associated with midostaurin, and gemtuzumab ozogamicin-related hepatotoxicity. Further research is essential to meet these unmet needs.

The impact of the polymorphism within genes in the microbiome-mediated drug response remains largely unexplored. The manually curated Pharmacogenomics Knowledgebase (PharmGKB) identifies key human genes involved in the individualized drug response; however, it does not incorporate microbial influences on drugs [162]. There are few studies revealing interactions of drugs with known pharmacogenetics with the human gut microbiome [163]. Developing models that capture the variation in the human microbiome can provide valuable insights into xenobiotic metabolism, particularly in the context of genetic variations and metabolic interactions between the host and associated microbes. This strategy may help predict individual drug responses and guide the design of effective, low-risk personalized treatments. The study of molecular pathological epidemiology (MPE) combines the disciplines of biology to provide research frameworks that link genetic and environmental factors, including pharmacological factors, to pathologic processes. The integration of microbiology into the MPE model (microbiology-MPE) can enhance our understanding of the complex interplay between the genetic background, environment, tumor cells, immune cells, drug response, and the microbiome in AML [164].

Recent advancements in high-throughput sequencing technologies, along with innovative statistical and computational methods for multi-omics data analysis, have enabled researchers to predict drug responses. However, the analysis of the omics data for effective treatment personalization in AML remains challenging due to the biological variability of the disease and data complexity. To address these challenges, researchers from the Karolinska Institute proposed MegaFun, a computational method designed to quantify the functional aspects of the microbiome using metagenomics data, to obtain information for prognostic models. The results of this study are still pending. The application of AI in microbiome data analysis is promising and will evolve into personalization in AML management shortly [165]. 

Besides the beneficial influence on the efficiency of drugs in cancer, some bacteria could promote chemotherapy resistance. Targeting specific bacterial species with bacteriophages offers a novel strategy to overcome chemoresistance. Achieving this goal requires identifying host strains with therapeutic properties to optimize phage combinations for selective pathobiont elimination.

Most prior studies have focused on GM. However, intratumor bacterial signatures may also play a role in regulating cancer development and progression, as documented in the tumor microbiome of pancreatic, lung, and breast cancers [166]. Fu et al. linked tumor-resident microbiota with cancer metastasis, suggesting that circulating tumor cells carrying intratumor bacteria may influence cancer cell survival [167]. In myeloid malignancies, the bone marrow and blood microbiome may play a key role. Advances in molecular technologies have introduced the concept of a blood microbiome, assessed in both healthy individuals and cancer patients. While a core healthy blood microbiome has not been established, dysbiosis appears relevant across various conditions [168,169]. The deep sequencing of blood and bone marrow from 1870 patients (including 612 with AML) revealed distinct dysbiosis patterns across four disease subtypes, with microbial profiles linked to gene mutations and myeloblast percentages. AML patients showed a higher bacterial burden, but lower diversity, dominated by *Proteobacteria*. Epstein–Barr virus presence correlated with a poorer prognosis in low-risk MDS. Blood and bone marrow samples showed comparable microbial profiles [170]. The blood microbiome exhibits potential for applications in risk stratification, diagnosis, monitoring, and drug development. Nonetheless, the investigation of the bone marrow and peripheral blood microbiome in the context of AML remains largely unexplored for these diagnostic and therapeutic objectives. Significant challenges include contamination in low-biomass samples and uncertainty regarding the viability of microbes identified through NGS. Predominantly, the circulating microbes are commensals and the disruption of physiological barriers in AML may facilitate their transient entry into the bloodstream. Typically, these microbes are cleared rapidly, lacking sustained colonization or functional significance [171]. Further investigation is necessary to distinguish between viable microbes and residual, inactive microbial DNA.

Currently, microbiome-related dietary modalities are classified as health products rather than pharmaceuticals. However, with ongoing technological and scientific advancements, these interventions may achieve the status of drugs with precise indications, dosages, and adverse effect profiles. Investigating their pharmacodynamics and pharmacokinetics within the context of the variable interindividual microbiome signature will present a considerable challenge for pharmacology with potential applications in hematology.

Based on previous evidence, it appears that shortly, microbiome management will become a core element of supportive care for cancer patients, similar to antiemetic, anticoagulant, and psychological support treatments. However, researchers still need to develop safe and effective strategies. Manipulating the composition of the microbial consortium for therapeutic purposes is one of the most explored topics within the field of pharmacomicrobiomics. In the case of AML, we are only beginning to understand the complex interactions between the microbiome and treatment, especially concerning new therapies.

## 7. Conclusions

Building on both foundational knowledge and current findings, we are the first to comprehensively position the microbiome within the therapeutic landscape of AML from multiple perspectives. The host microbiome and AML therapy are inter-related and affect each other through various mechanisms, including drug metabolism, immunomodulation, direct anticancer effects, and changes in microbial composition. Patients with AML constitute a distinct population where numerous additional factors significantly complicate these relationships. Common side effects, resistance to treatments, and the volatile bioavailability of drugs limit the efficacy of the standard therapy, a critical element for patient outcomes. The human microbiome plays a crucial role in orchestrating all these effects. A mechanistic and functional understanding of the relationships between the microbiome and therapy in AML patients is a clue to optimizing treatment and improving outcomes. It requires complementing microbiota characterization with a detailed analysis of the resistome, metagenome, and metabolome using novel techniques. While targeting human microbes offers promising anti-cancer potential, safety concerns remain a key limitation. The dynamic and multifaceted field of pharmacomicrobiomics represents a significant advancement toward personalized medicine; however, in AML, it remains in the early stages of development, highlighting the urgent need for ongoing progress.

## Figures and Tables

**Figure 1 biomedicines-13-01761-f001:**
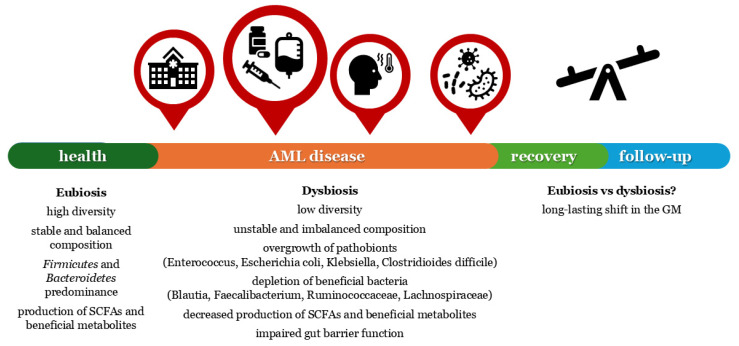
Critical timepoints of microbiome disruption and alteration during the clinical course of AML patients. Red circles indicate critical disruption points for the microbiome in AML patients where targeted therapeutic intervention and future research may be most beneficial (hospital administration, treatment, neutropenic fever, and infection). AML: acute myeloid leukemia; GM: gut microbiome; SCFAs: short-chain fatty acids.

**Figure 2 biomedicines-13-01761-f002:**
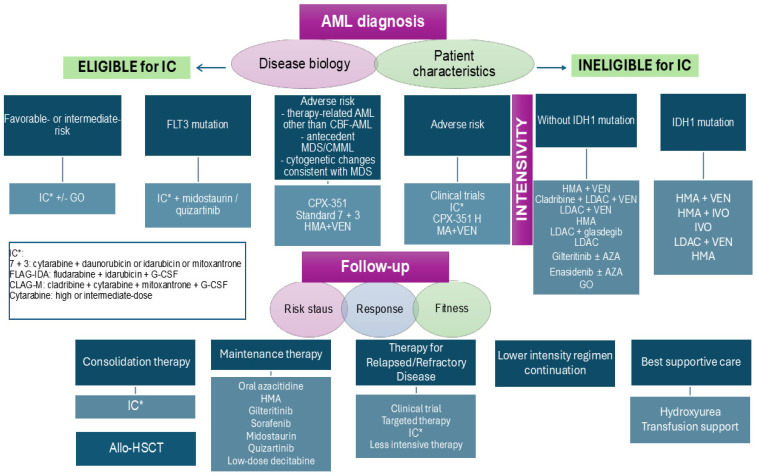
Treatment algorithm of AML according to NCCN 2025 recommendation. After initial disease biology and clinical evaluation, patients are stratified to intensive or lower-intensity therapy, followed by consolidation with/without HSCT and maintenance. Therapy options are presented. AML: acute myeloid leukemia; allo-HSCT: allogenic hematopoietic stem cell transplantation; CMML: chronic myelomonocytic leukemia; CPX-351: liposomal formulation of a fixed combination of daunorubicin and cytarabine; *FLT3*: FMS-like tyrosine kinase 3; G-CSF: granulocyte colony-stimulating factor; GO: gemtuzumab ozogamicin; HMA: hypomethylating agent; IC: intensive chemotherapy; *IDH*: isocitrate dehydrogenase; IVO: ivosidenib; LDAC: low-dose cytarabine; MDS: myelodysplastic syndrome; VEN: venetoclax.

**Figure 3 biomedicines-13-01761-f003:**
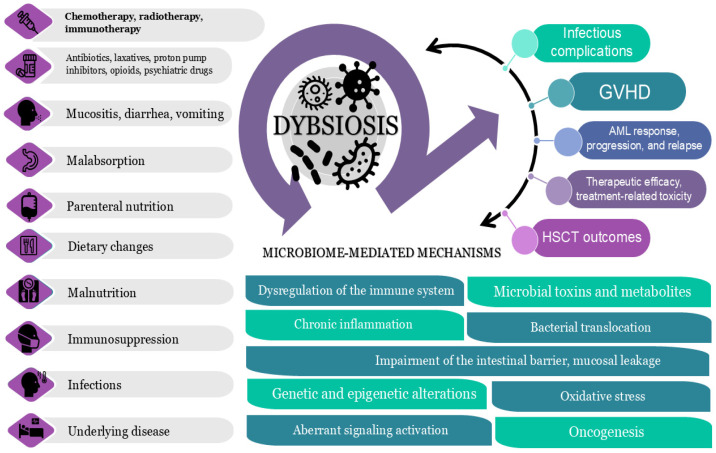
Interactions between microbiome alterations, AML treatment, and clinical outcomes. A comprehensive overview of the inter-relationships between microbiome-disrupting factors, AML treatment, resulting dysbiosis, and associated clinical outcomes, with an emphasis on potential underlying mechanisms. AML: acute myeloid leukemia; GVHD: graft-versus-host disease; HSCT: hematopoietic stem cell transplantation.

**Figure 4 biomedicines-13-01761-f004:**
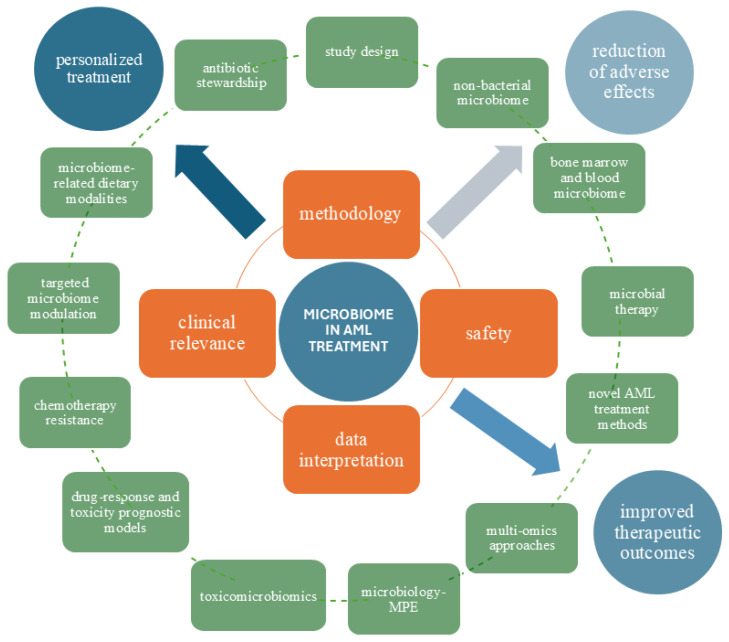
Current challenges, emerging directions, and future applications of the microbiome in AML treatment. Orange rectangles highlight key limitations—methodology: (1) variability in treatment regimens and drugs, (2) lack of adjustment for confounding factors such as diet and antibiotic use, (3) discrepancies in sequencing technologies and analytical pipelines, and (4) absence of standardized diagnostic criteria; safety: microbiome-based interventions in immunocompromised individuals; data interpretation: (1) high inter-individual and population variability and (2) uncertainty regarding microbial viability; and clinical relevance. Green rectangles outline key research needs and directions—study design: (1) rigorous experimental frameworks, (2) multi-spatial and temporal assessment, (3) in vitro and in vivo models, (4) longitudinal cohorts, (5) mechanistic investigations, and (6) clinical interventional trials; non-bacterial microbiome: (1) archaea, (2) fungi, and (3) viruses; bone marrow and blood microbiome; microbial therapy in AML: (1) immunotherapeutic applications utilizing live, attenuated, or genetically modified bacteria, either alone or with conventional treatments, (2) targeted delivery of anticancer agents, (3) bacterial expression of tumor-specific antigens, (4) delivery or expression of suppressor genes, anti-angiogenic genes, and suicide genes, (5) RNA interference mechanisms, and (6) activation of pro-drugs through bacterial cleavage; microbiome in novel AML treatment methods: (1) immunotherapy and (2) targeted therapies; toxicomicrobiomics: (1) neuropathy, (2) cardiotoxicity, (3) hepatotoxicity, and (4) mental disorder; microbiology-MPE; multi-omics approaches: (1) genomics, (2) epigenomics, (3) transcriptomics, (4) proteomics, and (5) metabolomics; drug–response and toxicity prognostic models; chemotherapy resistance; targeted microbiome modulation; microbiome-related dietary modalities; and antibiotic stewardship. Blue circles represent future prospects for microbiome integration in AML treatment: personalized treatment; improved therapeutic outcomes; reduction of adverse effects. Microbiology-MPE: microbiology–molecular pathological epidemiology.

**Table 1 biomedicines-13-01761-t001:** Medications for AML.

Group	Types	Drugs
Chemotherapy	Antimetabolites	Cytarabine
		Fludarabine
		Cladribine
		Clofarabine
		Thioguanine
		Decitabine
		Methotrexate
		Azacitidine
	alkylating agents	Cyclophosphamide
	anti-microtubule agents	vincristine
	topoisomerase inhibitors	doxorubicin
		mitoxantrone
	cytotoxic antibiotics	idarubicin
		daunorubicin
		mitoxantrone
		doxorubicin
targeted therapy	inhibitor of the b-cell lymphoma 2 regulator protein	venetoclax
	fms-related tyrosine kinase 3 inhibitors	midostaurin
		quizartinib
		gilteritinib
		sorafenib
	menin inhibitor	revumenib
	isocitrate dehydrogenase inhibitors	enasidenib
		ivosidenib
		olutasidenib
	antibody drug conjugate	gemtuzumab ozogamicin
	hedgehog pathway inhibitor	glasdegib
Other	hypomethylating agents	azacitidine
		decitabine
		dexamethasone
	Steroids	prednisone
	Retinoids	all-trans retinoic acid
		arsenic trioxide
		hydroxycarbamide
	combinations of antineoplastic agents	daunorubicin hydrochloride and cytarabine liposome
		cytarabine, daunorubicin hydrochloride, and etoposide phosphate

**Table 2 biomedicines-13-01761-t002:** Summary of studies related to the effect of antibiotics on the microbiome in AML.

Ref	Study Group	Sampling Time Point	Methodology (Material)	Antibiotics	Use	Comparator	Main Results			
							Alpha Diversity	Composition	Resistome	Metabolites
[46]	n = 94 HSCT recipients (AML n = 44)	serial analyses from before HSCT to 35 days post-HSCT	16S ribosomal RNA gene sequencing (fecal specimens)	vancomycin	P	temporal variations of microbiota profiles	↓	↔	n/a	n/a
				ciprofloxacin, levofloxacin	P	temporal variations of microbiota profiles	↓	10-fold ↓ in *Proteobacteria* dominance	n/a	n/a
				metronidazole	T	temporal variations of microbiota profiles	↓	3-fold ↑ in enterococcal dominance	n/a	n/a
				cephalosporins, beta-lactam–beta-lactamase combinations, carbapenems	T	temporal variations of microbiota profiles	↓	↔	n/a	n/a
[47]	n = 34 HSCT recipients (AML n = 14)	serial analyses from before HSCT to day 28 post-HSCT	next-generation sequencing (fecal specimens), strain-specific enterococcal PCR (fecal specimens), liquid chromatography–tandem mass spectrometry of urinary IS (urine)	trimethoprim/sulfamethoxazole followed by ciprofloxacin + metronidazole	P	temporal variations of microbiota profiles		↑ in the proportion of *E. faecalis* and *E. faecium*; ↓ in other *Firmicutes* and phyla	n/a	↓ 3-IS levels
				systemic antibiotics	T	temporal variations of microbiota profiles	↓	↑ mainly in *E. faecalis* and, to a lesser extent, *E. faecium*	n/a	↓ 3-IS levels
[48]	n = 857 HSCT recipients (AML n = 277)	prior to and after the initiation of a specific antibiotic treatment	16S ribosomal RNA gene sequencing (fecal specimens)	vancomycin + ciprofloxacin	P	comparison with other	n/a	minor perturbations	n/a	n/a
				iv or oral trimethoprim/sulfamethoxazole	T	comparison with other	n/a	minor perturbations	n/a	n/a
				oral atovaquone	T	comparison with other	n/a	minor perturbations	n/a	n/a
				imipenem–cilastatin	T	comparison with other	n/a	minor perturbations	n/a	n/a
				piperacillin–tazobactam	T	comparison with other	n/a	greater ↓ in *Bacteroidetes* and *Lactobacillus*; trend towards a ↓ in *Clostridia* and *Actinobacteria* (no statistical significance); ↓ in *Enterococcus*, *Akkermansia*, and *Erysipelotrichia* (similar to aztreonam and cefepime)	n/a	n/a
				aztreonam	T	comparison with other	n/a	↓ in *Enterococcus*, *Akkermansia*, and *Erysipelotrichia* (similar to piperacillin–tazobactam)	n/a	n/a
				cefepime	T	comparison with other	n/a	↓ in *Enterococcus*, *Akkermansia,* and *Erysipelotrichia* (similar to piperacillin–tazobactam)	n/a	n/a
[49]	n = 394 HSCT recipients (acute leukemia n = 219)	serial analyses from before conditioning and weekly within the first 28 days after HSCT	3-IS (urine), 16S ribosomal RNA gene sequencing (fecal specimens)	rifaximin	P	ciprofloxacin–metronidazole	n/a	↓ in *E. faecium* and *E. faecalis*	n/a	↑ 3-IS levels
[50]	n = 360 HSCT recipients (AML n = 197)	serial analyses from before HSCT to the time of stem cell engraftment	16S ribosomal RNA gene sequencing (fecal specimens), concentrations of SCFAs using targeted metabolomics methodology (fecal samples)	metronidazole	T	temporal variations of microbiota profiles	n/a	↓ in butyrate-producing bacteria	n/a	butyrate, acetate, propionate, and desamino-tyrosine correlated with the abundance of butyrate-producing bacteria
				beta-lactams	T	temporal variations of microbiota profiles	n/a	↓ in butyrate-producing bacteria	n/a	
				vancomycin	T	temporal variations of microbiota profiles	n/a	↔	n/a	
				fluoroquinolones	T	temporal variations of microbiota profiles	n/a	↔	n/a	
[51]	n = 60 (AML n = 26)	within 7 days after antibiotics exposure	16S ribosomal RNA gene sequencing (fecal specimens)	levofloxacin	P	no antibiotics	↑	trend toward ↓ dominance of non-*Bacteroidetes*; ↓ in *Proteobacteria*; ↑ in *Lachnospiraceae*, *Ruminococcaceae*, *Blautia*	n/a	n/a
				oral vancomycin	P	no antibiotics	↓	↓ in *Bacteroidetes;* ↑ risk of dominance of non-*Bacteroidetes*	n/a	n/a
				cefepime, piperacillin–tazobactam, meropenem	T	no antibiotics	↓	trend toward ↑ *Enterococcus*; ↓ in *Clostridia* and *Blautia*	n/a	n/a
[52]	n = 161 HSCT recipients (AML n = 87)	serial analyses within the first 10 days after HSCT	3-IS (urine), 16S ribosomal RNA gene sequencing (fecal specimens)	rifaximin	P	rifaximin with/withouTsystemic antibiotics vs. ciprofloxacin–metronidazole with/without systemic antibiotics	↑	↑ in *Clostridium* cluster XIVa (CCXIVa) abundance; ↓ in enterococcal load (not statistically significant)	n/a	↑ 3-IS levels
				ciprofloxacin + metronidazole	P	temporal variations of microbiota profiles	↓	domination of *Akkermansia*, *Eubacterium*, or *Enterococcus*	n/a	↓ 3-IS levels
				piperacillin–tazobactam, meropenem + vancomycin, ceftazidime, vancomycin	T	temporal variations of microbiota profiles	↓	domination of *Akkermansia*, *Eubacterium*, or *Enterococcus*	n/a	↓ 3-IS levels (except vancomycin alone)
				systemic antibiotics	T	temporal variations of microbiota profiles	↓	↑ mainly in *E. faecalis* and, to a lesser extent, *E. faecium*	n/a	↓ 3-IS levels
[53]	n = 8 HSCT recipients (AML n = 8)	serial analyses from before HSCT to 85 days post-HSCT	WGS metagenome sequencing (fecal specimens)	fluoroquinolones	P	temporal variations of the gut resistome in each individual	n/a	ARU26—↑ in *Bacteroides* sp. *D1*, *Prevotella intermedia*, *Capnocytophaga ochracea*, *and Bacteroides fragilis species*; ARU38—↑ in *B. fragilis* and *Bacteroides* sp.	↑ trend for AMR genes: ARU4 (tetracycline inhibitor), ARU26 (β-lactamase CFXA3), and ARU38 (erythromycin resistance); consolidation of AMR genes present before transplanting and acquisition of new AMR genes, particularly in aGvHD-positive patients, extending beyond the antibiotics used during treatment	n/a
				beta-lactams	T	temporal variations of the gut resistome in each individual	n/a			n/a
[54]	n = 97 AML	serial analyses from baseline to neutrophil recovery during induction chemotherapy	16S ribosomal RNA gene sequencing (oral swabs and fecal specimens)	carbapenem	T	temporal variations of microbiota profiles	↓ when carbapenems for >72 h	n/a	n/a	n/a
				cephalosporin	T	temporal variations of microbiota profiles	↔	n/a	n/a	n/a
				piperacillin–tazobactam	T	temporal variations of microbiota profiles	↔	n/a	n/a	n/a
[43]	n = 20 HSCT recipients (AML n = 10); n = 20 intensively treated acute leukemia (AML = 16)	acute leukemia—serial analyses from day 1 of chemotherapy until day 28 or discharge; HSCT—serial analyses from the day of transplantation until day 14 after transplantation.	16S ribosomal RNA gene sequencing (fecal specimens)	levofloxacin	P	comparison of variations of microbiota profiles in 2 cohorts of patients	↓ in both cohorts; n/a for particular antibiotics	↓ in *Enterococcus* domination (both groups)	n/a	n/a
				cephalosporins third-generation or higher	T	comparison of variations of microbiota profiles in 2 cohorts of patients	↓ in both cohorts; n/a for particular antibiotics	↑ in *Lactobacillus* domination (acute leukemia); ↑ in *Enterococcus* domination (acute leukemia)	n/a	n/a
				iv vancomycin	T	comparison of variations of microbiota profiles in 2 cohorts of patients	↓ in both cohorts; n/a for particular antibiotics	↑ in *Lactobacillus* domination (acute leukemia)	n/a	n/a
				oral vancomycin	T	comparison of variations of microbiota profiles in 2 cohorts of patients	↓ in both cohorts; n/a for particular antibiotics	no impact on domination	n/a	n/a
				piperacillin–tazobactam	T	comparison of variations of microbiota profiles in 2 cohorts of patients	↓ in both cohorts; n/a for particular antibiotics	no impact on domination	n/a	n/a
				carbapenems	T	comparison of variations of microbiota profiles in 2 cohorts of patients	↓ in both cohorts; n/a for particular antibiotics	no impact on domination	n/a	n/a
				metronidazol/clindamycin	T	comparison of variations of microbiota profiles in 2 cohorts of patients	↓ in both cohorts; n/a for particular antibiotics	↑ in *Enterococcus* domination (acute leukemia)	n/a	n/a
				linezolid/daptomycin	T	comparison of variations of microbiota profiles in 2 cohorts of patients	↓ in both cohorts; n/a for particular antibiotics	no impact on domination	n/a	n/a
				cefepime + iv vancomycin	T	comparison of variations of microbiota profiles in 2 cohorts of patients	↓ in both cohorts; n/a for particular antibiotics	no impact on domination	n/a	n/a
[55]	n = 708 HSCT recipients (AML n = 360)	serial analyses from the start of pretransplant conditioning until engraftment	16S ribosomal RNA gene sequencing (fecal specimens)	ciprofloxacin	P	no antibiotics	n/a	↓ in BSIs and intestinal colonization by Gram-negative microbes, including *Klebsiella*, *Citrobacter*, *Enterobacter*, and *Desulfovibrio*; ↑ in *Escherichia*, *Pseudomonas*, and *Stenotrophomonas*; ↑ in breakthrough with *E. coli* in intestinal colonization and BSIs	↑ in fluoroquinolone resistance	n/a

AMR: antimicrobial resistance. BSIs: bloodstream infections. IS: indoxyl sulfate. n/a: not applicable. P: prophylaxis. SCFA: short-chain fatty acids. T: treatment. WGS: whole-genome shotgun. ↓: decreased. ↔: stable. ↑: increased.

**Table 3 biomedicines-13-01761-t003:** Completed and ongoing clinical trials on the microbiome in AML.

ClinicalTrials.Gov ID	Purpose	Intervention	Outcome Measure	Study Population	Study Design	State	Results
NCT06899581	to assess the impact of leukemia treatment on GM and its recovery trajectory	dietary supplement: enteral nutrition food-derived ingredient	GM, alpha and beta-diversity, *Firmicutes* to *Bacteroidetes* ratio, composition	Child	observational, case–control, prospective	Recruiting	no results posted
NCT02928523 (ODYSSEE)	to assess autologous FMT efficacy in preventing dysbiosis complications in AML patients receiving intensive treatment	autologous FMT MaaT011 post-chemotherapy	dysbiosis correction (microbiota alpha and beta diversity), MDRB eradication (bacterial culture), biological parameters	Adult	interventional, phase 1 and phase 2, single group assignment, open label	completed	restoration of GM diversity and community recovery of microbial richness and diversity to baseline levels safety
NCT03959241	to assess whether GM diversity at neutrophil engraftment predicts one-year non-relapse mortality in patients receiving reduced-intensity allo-HSCT	tacrolimus/methotrexate versus post-transplant cyclophosphamide/tacrolimus/mycophenolate mofetil in non-myeloablative/reduced intensity conditioning allogeneic peripheral blood stem cell transplantation	percentage of participants with GVHD/relapse or progression-free survival at one year	Adult	interventional, phase 3, randomized, parallel assignment 1:1, open label	completed	no results on GM finding posted
NCT02949427	to characterize oral and nasal microbiota, including fungi, before and after chemotherapy, and supportive care	chemotherapy and supportive care	diversity index of oronasal mycobiome and microbiome, relative abundance of the oronasal fungal microbiome and microbiome	4 years to 21 years	observational, cohort, prospective	completed	no results posted
NCT04940468	to determine whether dietary intervention to increase fiber and decrease fat reduces *C. difficile* infection recurrence in a cohort of oncology patients	diet higher in fiber and lower in fat	*C. difficile* toxins A and B, fecal microbiome (16S rRNA, shotgun metagenomic sequencing)	9 years and older	interventional, randomized, parallel assignment, open label	recruiting	no results posted
NCT06355583	to test the ability to restore GM to healthier levels in patients with blood cancers scheduled to have HSCT	capsule with communities of dried, intestinal microorganisms from screened, pooled human stool samples (IMT) swallowed 2 weeks before HSCT	tolerability and acceptability of IMT, GM diversity (richness and evenness), health, infective/microbiological, and hematological outcomes (days of fever, admission to intensive care unit, survival, non-relapsed mortality, and incidence of GVHD)	18 years and older	interventional, phase 2, randomized, parallel assignment, masking: triple (participant, care provider, investigator)	recruiting	no results posted
NCT04214249	to correlate microbial/metabolome changes at baseline and changes with clinical response (immune-checkpoint expression, kinetics of immune cell subset recovery, and programming) in the standard of care and experimental arm	pembrolizumab in combination with conventional intensive chemotherapy as frontline therapy in patients with AML	rate of MRD negative—complete response/complete remission with incomplete recovery, immune cell subsets, PD-1 and PD-L1 expression, protein signatures, T cell receptor sequencing, GM	18 years to 75 years	interventional, phase 2, randomized, parallel assignment, open label	active, not recruiting	no results posted
NCT05596981	to investigate the effect of sorafenib maintenance therapy in FLT3-ITD-positive AML patients after allo-HSCT on GM	sorafenib	variation of GM composition and diversity (16s rRNA sequencing of serial stool samples), variation of gut barrier integrity (serum levels of zonulin, I-FABP, and citrulline or other potential candidates), treatment outcomes, GVHD	18 years to 65 years	observational, cohort, perspective	recruiting	no results posted
NCT03678493	to asses efficacy of FMT in AML patients and allo-HSCT recipients	3 treatments of oral capsule FMT vs. placebo after each exposure to antibacterial antibiotics	number of infections, FMT engraftment, GVHD	18 years and older	interventional, randomized, double-blind, placebo-controlled, open-label	completed	safety, intestinal dysbiosis amelioration, no decrease in infections
NCT04629430	to see whether HSCT patients can consistently eat a diet rich in prebiotics	prebiotic foods/drinks	frequency of participants ingesting the required diet, GVHD, incidence of *C. difficile* infection, patient weight, number of days to neutrophil engraftment	18 years and older	interventional, single group assignment, open label	completed	no results posted

allo-HSCT, allogenic hematopoietic stem cell transplant; FMT, fecal microbiota transplantation; GM, gut microbiota; GVHD, graft-versus-host disease; MDRB, multidrug-resistant bacteria; MRD, minimal residual disease.

## Data Availability

Not applicable.

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
