# Peer review of "Microbial Crosstalk with Therapy: Pharmacomicrobiomics in AML—One Step Closer to Personalized Medicine"

_biomedicines, 2025, doi:10.3390/biomedicines13071761_

Round 1

Reviewer 1 Report

Comments and Suggestions for Authors

Recommendations:

  1. Percent match is a bit high. reduce it, please
  2. The issue with this review is that a a high proportion of the study is focused on antibiotic treatment influence on microbiome, but with no corelation with the impact on AML treatment response.
  3. No discussion on impact of microbiome on Graft vs. Host disease (GvHD)  see this: https://doi.org/10.3390/children12020166
  4. The manuscript is rather monotonous. you should add some constructive figures.
  5. I did not see any studies on the CDI infection in AML and microbiome modulation in this review. 
Comments on the Quality of English Language

Minor revisions required

Author Response

Thank you for your valuable comment. In response, we have thoroughly revised the manuscript to enhance its clarity, readability, and overall scientific tone. Specifically, we have:

  • Streamlined overly long or complex sentences to improve flow and comprehension;

  • Removed redundant phrasing and repeated concepts to maintain focus and precision;

  • Actively minimized the use of passive voice in favor of more direct and engaging sentence constructions, where appropriate.

We believe these revisions have significantly improved the manuscript’s readability and presentation. All changes are reflected throughout the revised text.

Reviewer 2 Report

Comments and Suggestions for Authors

The current literature review pharmacomicrobiomics in acute myeloid leukemia is comprehensive in my opinion.

Table 2 is not presented completely in the PDF file. Please double check table 2. Also, a figure summarizing mechanisms of how AML therapies influence dysbiiosis should be made. 

Author Response

We sincerely thank you for your positive evaluation and constructive feedback. We greatly appreciate your thoughtful review, which supports the value and relevance of our work. Your comments have helped us refine the manuscript further, and we are grateful for the opportunity to improve its clarity and scientific rigor.

Reviewer 3 Report

Comments and Suggestions for Authors
  1. Is there any mechanism shows how microbiome is reponsible for the pathogensis of AML.
  2. Is microbiome influence the formation of blast cells in AML.
  3. How microbiome interact with the oncogenic signaling pathways in AML.
  4. Discuss the relationship of microbiome with the molecualr, cytogenetics profile of AML if any.
  5. Discuss the list results of completed, ongoing clinical trails on microbiome in AML, if possible.

Author Response

(The authors gave the same response as above.)

Reviewer 4 Report

Comments and Suggestions for Authors

The manuscript entitled “Microbial crosstalk with therapy: pharmacomicrobiomics in AML-one step closer to personalized medicine” by Nowicka et al. addresses a timely, clinically relevant and under-explored topic in a well-articulated manner. This review manuscript is thorough, and authors have successfully explained the existing knowledge gap in current microbiome research in AML. However, I think with minor structural refinement, improved clarity and reduction of redundancy at some places might enhance the quality of the manuscript even more. In the attached word doc file I have provided the details of the areas of improvement of the manuscript.

Comments on the Quality of English Language

Sentences are too long. Shorter meaningful sentences and consistent usage of active voice would be necessary for better clarity. Grammatical and spelling errors can be corrected using commercial language modification software like "Grammarly".

Author Response

(The authors gave the same response as above.)

Round 2

Reviewer 1 Report

Comments and Suggestions for Authors

Recommendations:

 Not fully addressed the comments! 

Comments on the Quality of English Language

English fine!

Round 3

Reviewer 1 Report

Comments and Suggestions for Authors

Congratulations to the authors!